# The Potential of a New Natural Vessel Source: Decellularized Intercostal Arteries as Sufficiently Long Small-Diameter Vascular Grafts

**DOI:** 10.3390/bioengineering11070700

**Published:** 2024-07-10

**Authors:** Yuan Xia, Haiyun Zhou, Jing-Song Ou, Yunqi Liu

**Affiliations:** 1Division of Cardiac Surgery, Cardiovascular Diseases Institute, The First Affiliated Hospital, Sun Yat-Sen University, Guangzhou 510080, China; xiay87@mail2.sysu.edu.cn; 2Department of Cardiac Surgery, The First Affiliated Hospital, Guangzhou Medical University, Guangzhou 510160, China; 3National-Guangdong Joint Engineering Laboratory for Diagnosis and Treatment of Vascular Diseases, NHC Key Laboratory of Assisted Circulation and Vascular Diseases (Sun Yat-Sen University), Key Laboratory of Assisted Circulation and Vascular Diseases, Chinese Academy of Medical Sciences, Guangdong Provincial Engineering and Technology Center for Diagnosis and Treatment of Vascular Diseases, Guangzhou 510080, China; 4Guangdong Provincial Key Laboratory of Brain Function and Disease, Zhongshan School of Medicine, Sun Yat-Sen University, Guangzhou 510080, China

**Keywords:** small-diameter vascular grafts (SDVGs), decellularization, intercostal artery, coronary artery bypass grafting (CABG), tissue engineering

## Abstract

Small-diameter vascular grafts (SDVGs) are severely lacking in clinical settings. Therefore, our study investigates a new source of biological vessels—bovine and porcine decellularized intercostal arteries (DIAs)—as potential SDVGs. We utilized a combination of SDS and Triton X-100 to perfuse the DIAs, establishing two different time protocols. The results show that perfusing with 1% concentrations of each decellularizing agent for 48 h yields DIAs with excellent biocompatibility and mechanical properties. The porcine decellularized intercostal arteries (PDIAs) we obtained had a length of approximately 14 cm and a diameter of about 1.5 mm, while the bovine decellularized intercostal arteries (BDIAs) were about 29 cm long with a diameter of approximately 2.2 mm. Although the lengths and diameters of both the PDIAs and BDIAs are suited for coronary artery bypass grafting (CABG), as the typical diameter of autologous arteries used in CABG is about 2 mm and the grafts required are at least 10 cm long, our research indicates that BDIAs possess more ideal mechanical characteristics for CABG than PDIAs, showing significant potential. Further enhancements may be necessary to address their limited hemocompatibility.

## 1. Introduction

In coronary artery bypass grafting (CABG), small-diameter vascular grafts (SDVGs) are typically required to restore normal blood supply to cardiac muscle, depending on the severity of the coronary artery disease (CAD) and the number of vessels involved, which usually entails the creation of one to three different grafts. The current gold standard remains the use of autologous arterial or venous grafts from the patient, as these do not trigger immunological reactions and exhibit high long-term patency rates [1,2,3]. However, this method has limitations. For instance, conventional bypass surgery often involves the use of the great saphenous vein and the internal thoracic artery. Although the internal thoracic artery is the preferred conduit in coronary artery bypass grafting due to its superior long-term patency and outcomes, concerns about its potential impact on sternum healing must be carefully considered [4,5]. The use of the great saphenous vein involves longer incisions, which can lead to wound infections, scarring, prolonged healing times, severe and enduring postoperative pain, and an increased likelihood of chronic pain and edema in the lower limbs. Furthermore, as a venous graft, it has a lower long-term patency compared with arterial grafts [6,7].

Although widely used in large and medium diameters in clinical settings, mature products with small diameters have yet to be used for synthetic vascular grafts due to their tendency to form thrombi and stenosis. An alternative treatment using tissue-engineered vascular grafts with excellent biocompatibility may potentially overcome these limitations. Nevertheless, these applications generally require high technical expertise and are costly and time-consuming [8,9,10,11].

Recent studies indicate that naturally derived decellularized tissues possess the structural and mechanical properties to support host cell migration and tissue regeneration [12,13,14]. Crucially, the decellularization process significantly reduces immunogenicity and allows for storage. The advantages of decellularized natural materials stem from their extracellular matrix structure and inherent bioactivity, offering superior benefits over synthetic materials. However, while decellularized vessels from various sources have been explored, they are typically less than 10 cm in length, which falls short of the requirements for CABG, and their diameters do not adequately match those of adult coronary arteries. Thus, there is an urgent need to identify more suitable natural sources for decellularized vessels in CABG.

In this study, for the first time, porcine intercostal arteries (PIAs) and bovine intercostal arteries (BIAs) were selected as bases for decellularized vessel preparation. Firstly, because an artery is present between each pair of ribs, a single animal can provide up to 22 intercostal arteries, making them abundantly available. Secondly, these arteries are of adequate length for CABG: the great saphenous vein typically measures 15–20 cm and the radial artery 10–15 cm, whereas the PIA measures approximately 14 cm and the BIA about 29 cm in length. Their diameters also match those used in CABG more closely; the internal thoracic artery typically measures 1–2 mm in diameter, and the radial artery measures 2–3 mm, while the PIA is about 1.5 mm and the BIA about 2.2 mm in diameter [15,16].

This study aims to explore the potential of PIAs and BIAs as SDVGs in CABG. Specifically, we first developed an effective decellularization protocol, followed by a series of experiments to validate the efficacy and safety of these animal-derived intercostal arteries. The findings of this study provide significant scientific evidence for vascular graft selection in CABG and may guide new directions in future cardiac surgical procedures [17,18,19,20,21,22]. 

## 2. Materials and Methods

### 2.1. Materials

The materials and reagents used in this study include phosphate-buffered saline (PBS, Sigma-Aldrich, St. Louis, MO, USA), penicillin–streptomycin (Life Technologies, Carlsbad, CA, USA), 2-[4-(2,4,4-trimethylpentan-2-yl) phenoxy] ethanol (Triton X-100, Sigma Aldrich, St. Louis, MO, USA), sodium dodecyl sulfate (SDS, Sigma Aldrich, St. Louis, MO, USA), and 4% paraformaldehyde (Biosharp, Hefei, China). For histological analysis, the reagents used for H&E staining, Masson’s trichrome staining, and Sirius Red staining were all purchased from Servicebio, Wuhan, China. 

### 2.2. Harvest and Decellularization of Intercostal Artery

Intercostal arteries were sourced from male Bos Taurus Angus (350–400 kg) and Sus scrofa domesticus (90–110 kg) at a slaughterhouse and extracted during rib sectioning. Following extraction, the arteries were tied off to close off any branches and immediately placed in cold phosphate-buffered saline that included 100 U/mL of penicillin and 100 g/L of streptomycin (P/S PBS), and then transported to our lab. Saline was used to flush out any remaining blood clots from the arteries.

For decellularization, we used a blunt 18-gauge needle, inserted into the vascular lumen, and secured it with surgical sutures. The volume of the solution used was 500 mL, with a perfusion pressure of approximately 70 mmHg. The arteries were perfused with two detergents sequentially at a flow rate of 50 mL/min: firstly with 1% *w*/*v* Triton X-100 and subsequently with 1% *w*/*v* SDS. This treatment was applied for either 24×2 or 48×2 h. Following each detergent treatment, the arteries were thoroughly rinsed with distilled water for 24 h to eliminate any detergent residues. Post-rinsing, the arteries were stored in P/S PBS at a temperature of 4 °C.

### 2.3. DNA Content and Characterization of Tissue

Residual DNA was quantified using a commercial DNA kit (Servicebio, Wuhan, China) to assess the effectiveness of the decellularization process. For histological analyses, DIAs were first fixed in 4% paraformaldehyde (Servicebio, Wuhan, China) for 24 h and then dehydrated through a graded series of alcohol (75%, 85%, 95%, and 100%). The samples were subsequently embedded in paraffin and sectioned into 5 μm thick slices using a microtome. After deparaffinization with xylene, the sections were stained with Hematoxylin and Eosin (H&E, Servicebio, Wuhan, China) to visualize their general tissue structure and with DAPI (Beyotime Biotechnology, Shanghai, China) for nuclear staining. Collagen distribution was identified using Masson’s trichrome (Servicebio, Wuhan, China) and Sirius Red staining (Servicebio, Wuhan, China). All histologically stained specimens were examined under an optical microscope (NIKON ECLIPSE E100, Tokyo, Japan) to analyze the tissue architecture and the effectiveness of decellularization.

In this study, the native groups and the completely decellularized groups (porcine 48 h and bovine 48 h) are collectively referred to as DIAs.

### 2.4. Scanning Electron Microscopy (SEM)

Lyophilized samples were sliced to reveal their inner surfaces and mounted on SEM stubs using conductive adhesive tape. They were then evenly coated with platinum in an argon atmosphere utilizing a sputter coater (ISC150, SuPro, Shenzhen, China). Observations were made with a scanning electron microscope (SU8010, HITACHI, Tokyo, Japan) set to 10 kV. Images were taken at 400× magnification across five to seven randomly chosen fields and were processed using Image J software (win64, NIH, Bethesda, MD, USA).

### 2.5. Mechanical Characterizations

The tensile strength of the scaffolds was evaluated using a universal mechanical tester (WD-5A, ZYYD, Shenzhen, China), securing the samples between grips set about 10 mm apart and stretching them longitudinally at a rate of 10 mm/min until they broke. To prevent drying, samples were kept moist during testing. Stress–strain relationships were captured using the tester’s associated software and further analyzed with Origin software (2022, OriginLab, Northampton, MA, USA).

For burst pressure tests, scaffolds were trimmed to lengths of 0.5 cm. Each piece was affixed to a pressure transducer (YK-100B, Yunyi, XiAn, China) at one end, which also featured a three-way valve, and the opposite end was tightly sealed. The scaffold was filled with saline, and pressure was incrementally increased via the valve until the scaffold ruptured. Peak burst pressures were read using SHILEK software (yk100B120B, win64, Yunyi, XiAn, China) and recorded in an Excel spreadsheet for subsequent analysis.

### 2.6. Cytocompatibility

The DIAs were sterilized using UV light for 48 h and then sectioned into 1 cm pieces and soaked in PBS for two days to prepare extract solutions. Subsequently, Human Umbilical Vein Endothelial Cells (HUVECs, American Type Culture Collection, Manassas, VA, USA) were plated in 96-well plates at about 2000 cells per well using endothelial growth medium (211–500, Sigma-Aldrich, St. Louis, MO, USA). These were incubated at 37 °C in a 5% CO_2_ atmosphere for 48 h. After this incubation, 10 μL of the scaffold extract was introduced to 100 μL of the medium in each well and incubated for an additional 24 h. Control wells were set up with the same cell density but without a scaffold extract. The cell viability was assessed by adding a CCK8 reagent (Sigma-Aldrich, St. Louis, MO, USA) and measuring the absorbance at 450 nm after 3 h of incubation using a Multiskan Mk3 microplate reader (Thermo Scientific, Waltham, MA, USA).

### 2.7. Hemocompatibility

We evaluated the hemolytic performance of materials by applying diluted fresh rabbit blood (containing 10% anticoagulant citrate dextrose), diluted in a 1:9 ratio with PBS, to a cut surface area (inner side) of 1 square centimeter. The negative control consisted of blood diluent that did not have any contact with the sample. After incubating the solution at 37 °C for 30 min, we aspirated it, centrifuged it at 3000 rpm for 3 min to separate the plasma and red blood cells, and then transferred the supernatant to a 96-well plate and measured the absorbance at 540 nm (O.D.540) to quantify free hemoglobin.

Similarly, to assess the coagulation properties, we added 1 mL of fresh rabbit blood to the same cut surface area and incubated it at 37 °C for 30 min. We then transferred the unclotted blood, added distilled water to lyse the red blood cells, and used a similar method as in the hemolysis evaluation to determine the amount of free hemoglobin.

We also separated platelet-rich plasma from fresh rabbit blood using two consecutive centrifugation steps: first at 250× *g* for 5 min, followed by 1800× *g* for 20 min. The platelets were then resuspended in an equal volume of PBS, and 1 mL of this platelet solution was applied to each sample’s cut surface area (inner side) and incubated at 37 °C for 30 min. The solution was then transferred to a new 96-well plate. To assess the quantity of remaining unaggregated platelets, we lysed the remaining platelets by adding distilled water and then measured the lactate dehydrogenase (LDH) activity using an LDH activity assay kit (Elabscience, Wuhan, China) to assess the remaining platelet quantity.

### 2.8. Statistical Analysis

All tests were statistically evaluated using Student’s *t*-test or one-way ANOVA test using GraphPad Prism 8 software, with a minimum *p* < 0.05. Unless otherwise stated, all relevant observations were repeated at least three times.

## 3. Results

### 3.1. The Diameters and Lengths of PIAs and BIAs Meet the Requirements for CABG

A macroscopic examination revealed the dimensions of the PIAs and BIAs (Figure 1A,B). The PIAs measured an average length of 14.67 ± 4.191 cm (Figure 1D) and had an internal diameter of 1.494 ± 0.3377 mm (Figure 1C). Conversely, the BIAs showed an average length of 29.25 ± 5.948 cm (Figure 1D) with a lumen diameter of 2.239 ± 0.2099 mm (Figure 1C). Since no noticeable differences were found in the length and diameter of the arteries pre- and post-decellularization, the statistical analysis was limited to PIAs and BIAs.

### 3.2. 48*2 h Protocol Can Yield DIAs with Low Immunogenicity

Decellularization was carried out using a perfusion device (Figure 2C). DAPI staining initially showed clearly visible cell nuclei in the PIAs and BIAs (Figure 2A). Treatment with SDS (24 h) and Triton X-100 (24 h) resulted in the partial removal of nuclear components. However, extending the treatment to 48 h with SDS and Triton X-100 resulted in almost complete elimination of nuclear materials. The DNA content in the PIAs and BIAs decreased following decellularization, with a reduction of approximately 60–70% after the 24 h*2 protocol, and it dropped below the international threshold of 50 ng/mg only after the 48 h*2 protocol (PIAs: initial 395.8 ± 53.96 ng/mg; post 24 h 167.9 ± 20.14 ng/mg; and post 48 h 26.84 ± 2.640 ng/mg; BIAs: initial 414.6 ± 79.47 ng/mg; post 24 h 144.5 ± 21.64 ng/mg; and post 48 h 25.48 ng/mg). For further experiments, only the native and fully decellularized intercostal artery groups, namely the PDIAs and BDIAs, were included for comparison.

### 3.3. Tissue Staining Indicated That the Collagen Fibers Were Largely Preserved

The scaffolds were analyzed before and after decellularization using three different staining techniques (Figure 3). H&E staining initially displayed visible cellular components and nuclei in both the PIAs (a) and BIAs (g). Post-decellularization, H&E staining revealed no remaining cells in either the PDIAs (d) or BDIAs (j), confirming the removal of nuclei from the vascular matrix, as supported by the DAPI staining results (Figure 2A). Meanwhile, the external elastic lamina was preserved in both the PDIAs and BDIAs, and the adventitia became notably loose (d,j). Masson’s trichrome staining results indicated that collagen fibers (colored blue) remained intact without notable damage, while smooth muscle fibers (dark red) were largely eliminated (b,e,h,k). This observation was supported by the results of the Sirius Red staining (c,f,i,l), which highlighted the preservation of collagen fibers (collagen type I: red, collagen type III: yellow). However, collagen type III fibers were removed in the PDIAs. The decellularization process yielded comparable results for both bovine and porcine vascular scaffolds.

### 3.4. Surface Morphology

Scanning electron microscopy (SEM) was used to examine the microstructure of the samples (Figure 4). Prior to decellularization, the adventitia of the PIAs was dense (a), but became loose after decellularization (b). Prominent intimal folds were visible in the PIAs before decellularization (c), but these folds were largely absent after decellularization (d). The changes observed in the BIAs to BDIAs (e, f) were consistent with those observed in the PIAs to PDIAs (g, h).

### 3.5. PDIAs and BDIAs Maintain Favorable Mechanical Properties

No substantial alterations were observed in the maximum stress (Figure 5A) and elongation at break (maximum strain) (Figure 5B) when comparing the PIAs with PDIAs and the BIAs with BDIAs. However, the performance of the BIAs and BDIAs was notably superior to that of the PIAs and PDIAs. Following decellularization, the burst pressure of the PDIAs saw a reduction (from 413.7 ± 37.79 to 368.5 ± 17.00 mmHg), whereas it increased for the BDIAs when compared with BIAs (from 1715 ± 16.89 to 1862 ± 22.93 mmHg). The burst pressure performance of the bovine vessels was significantly more robust than that of the porcine vessels (Figure 5C).

### 3.6. Adequate Cellular Compatibility Yet Suboptimal Hemocompatibility for BDIAs and PDIAs

To evaluate the biocompatibility of the scaffolds, extracts from the PIAs, PDIAs, BIAs, and BDIAs were integrated into the culture medium of pre-cultured HUVECs. Following a 24 h incubation, their cell viability was assessed using the CCK8 assay kit. The findings revealed no significant variations in the biological activity across all scaffold groups (Figure 6A), indicating generally favorable biocompatibility.

During the blood compatibility assessment, the PDIAs and BDIAs performed worse than the negative control group in the coagulation test (Figure 6C). In terms of hemolytic performance, although the decellularized vessels exhibited better anti-hemolytic properties (PIAs: 36.7 ± 2.066%, PDIAs: 24.86 ± 5.151%, BIAs: 36.50 ± 1.380%, and BDIAs: 29.27 ± 8.163%), they still failed to meet the international standards for anti-hemolytic performance of biomaterials (<5%). There were no significant differences in anti-platelet aggregation between these decellularized vessels and the negative control (Figure 6B). These observations suggest that the decellularization process may have effectively removed substances that promote platelet aggregation and hemolysis from the vessels. However, it may also have depleted factors that inhibit coagulation, thereby impacting the hemocompatibility of the scaffolds.

## 4. Discussion

CAD remains a major threat to human health and has consistently been a focus of research. In clinical practice, for patients with severe arterial narrowing, aside from pharmacotherapy and stent placement, CABG is considered the gold standard of treatment. However, the use of autologous vessels, which are commonly employed in these surgeries, has its limitations as it can cause additional physical trauma. This underscores the urgent need to develop artificial arterial substitutes. When synthetic vascular grafts failed to meet the requirements for small-diameter vascular grafts, researchers turned their attention to decellularized natural vessels. However, the natural vessels selected in other studies did not possess suitable geometric characteristics for CABG.

Therefore, to identify suitable SDVGs for CABG, this study explored the potential of bovine and porcine decellularized intercostal arteries as materials [23,24]. Our findings indicate that BDIAs, due to their longer length, wider diameter, and superior mechanical properties, show greater potential for use in CABG compared with PDIAs [25].

Traditional decellularization methods using orbital shakers have limitations [15,26,27,28,29,30], primarily because the detergents do not sufficiently contact the inner walls of the vessels, leading to incomplete cell removal. To overcome this, we adapted a perfusion method using a peristaltic pump to ensure comprehensive contact between the vascular inner walls and the decellularization agents, inspired by other studies [11,31,32,33]. Initial protocols from these studies [11], which suggested a decellularization duration of 5 h [34], were inadequate for complete cell removal in BIAs and PIAs. Consequently, we extended the duration to 48 h for both SDS and Triton X-100, which achieved thorough decellularization (Figure 2) [11,35]. Additionally, our research found that the rate of perfusion did not significantly impact the effectiveness of decellularization [36]. We conducted the perfusion of vessels at varying flow rates of 10 mL, 20 mL, 50 mL, 100 mL, 150 mL, and 250 mL per hour, observing residual nuclei via DAPI staining (Appendix A). The DNA quantification revealed no significant variances between the groups (Appendix A). To keep the vessels expanded and ensure thorough contact with the decellularizing agents during the decellularization process, and to avoid a quick loss of mechanical integrity, a flow rate of 50 mL/min was selected as optimal for the procedure.

The length and diameter of BIAs provide the capacity to create grafts that are larger in diameter or longer, which is essential for CABG. In CABG procedures, the quantity of grafts required often varies based on the specific health conditions of the patient. Each segment of these vascular grafts has an average length of about 29 cm, which comfortably surpasses the typical minimum requirement of 10+ cm needed for CABG, thus allowing for more surgical flexibility. According to studies, the ideal diameter for these grafts should be close to that of the coronary arteries, which generally measure 2–3 mm in adults. The approximately 2.2 mm diameter of BDIAs is thus well matched for this application. The robust mechanical properties and larger diameters of BDIAs position them as excellent options for maintaining consistent long-term blood flow, avoiding the significant degradation or constriction that is often observed in grafts with inferior mechanical characteristics. This enhancement could substantially increase the long-term patency rates of grafts, potentially reducing the need for reoperation due to graft failures. Additionally, their burst pressure of around 1800 mmHg is well suited to meeting arterial physiological requirements [37]. PDIAs, with their inferior mechanical properties, might be better suited for other types of surgeries that have reduced mechanical strength requirements.

This study underscores the promising use of DIAs in vascular grafting but acknowledges certain limitations. For example, while perfusion rates did not markedly influence the decellularization results, the ideal perfusion settings might differ across various vessel sizes. Additionally, the enduring impacts on the vascular matrix after decellularization and the hemocompatibility of the vessels warrant more extensive study. Given the deficiency in hemocompatibility of the vessel interiors, future work will focus on surface modification techniques to improve this characteristic (Appendix A). Potential enhancements might involve applying hydrophilic coatings or incorporating antithrombotic agents to improve the hemocompatibility and endothelialization of these grafts, thus aligning their functionality and performance closer to that of native vessels. Due to the noted potential decrease in hemocompatibility within the vessel walls, it is crucial to develop advanced surface modification strategies to counteract this issue. Future investigations may look into employing hydrophilic polymeric coatings, such as polyethylene glycol (PEG) or phosphorylcholine, which have proven effective in significantly reducing protein adsorption and platelet adhesion, thereby boosting blood compatibility [38,39]. Furthermore, incorporating antithrombotic substances, such as heparin, or agents that enhance endothelial cell growth, such as vascular endothelial growth factor (VEGF), might facilitate endothelialization and replicate the antithrombotic functions of natural vessels [40]. These adjustments are designed to enhance hemocompatibility and aid in the endothelialization process, thereby boosting the overall function and effectiveness of the grafts to better resemble natural arteries. These strategic improvements could establish these engineered grafts as sustainable alternatives for vascular surgeries over the long term. Moreover, to decrease the duration needed for decellularization while maintaining effective cell elimination and preserving both mechanical and biocompatible qualities, subsequent research may investigate the use of higher concentrations or varied types of decellularizing agents.

## 5. Conclusions

In this study, we demonstrated the potential of using decellularized intercostal arteries (DIAs) for vascular grafting, marking the first discovery of natural vessel sources that are more suitable for coronary artery bypass grafting (CABG). The length, diameter, and mechanical properties of bovine DIAs (BDIAs) perfectly match the requirements for CABG, while porcine DIAs (PDIAs) might be more suitable for other types of surgeries due to their weaker mechanical performance. In future studies, we will further process BDIAs and PDIAs to address their hemocompatibility deficiencies, aiming for potential clinical applications in tissue engineering and regeneration.

## Figures and Tables

**Figure 1 bioengineering-11-00700-f001:**
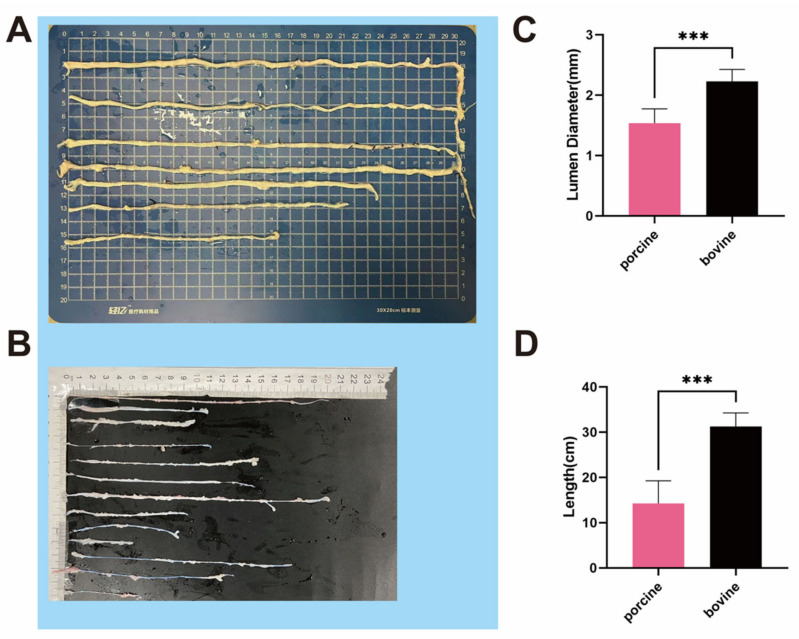
Macroscopic images of blood vessels and statistical analysis. (**A**) BIAs (bovine intercostal arteries); (**B**) PIAs (porcine intercostal arteries); (**C**) statistical analysis of the lumen diameter of BIAs and PIAs; and (**D**) statistical analysis of the lengths of BIAs and PIAs. n ≥ 6, *** indicates *p* < 0.001.

**Figure 2 bioengineering-11-00700-f002:**
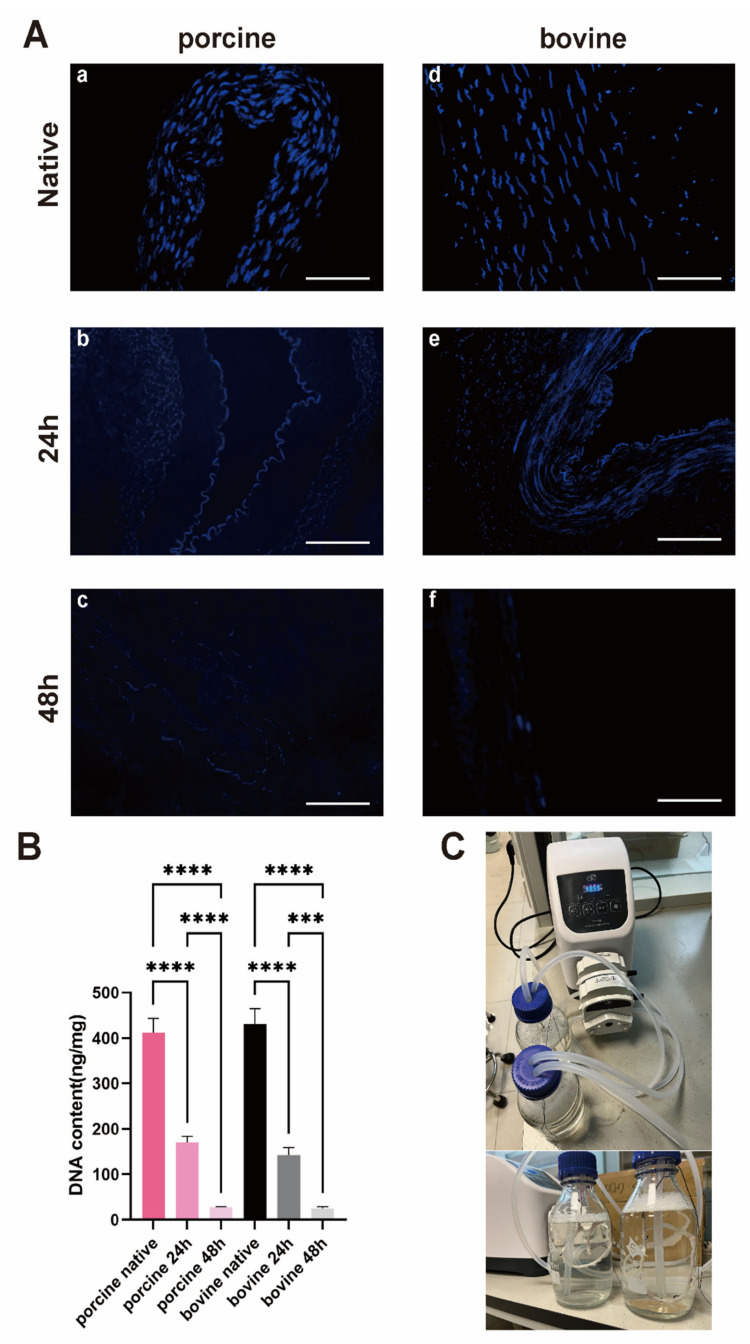
Evaluation of decellularization effectiveness. (**A**) DAPI staining of bovine and porcine intercostal arteries: porcine group (**a**,**b**,**c**); bovine group (**d**,**e**,**f**). Scale bar is 100 μm. (**B**) DNA content for each group. Test results for each animal are presented as native, 24 h decellularization group, and 48 h decellularization group. *** indicates *p* < 0.001, **** indicates *p* < 0.0001. (**C**) Perfusion device.

**Figure 3 bioengineering-11-00700-f003:**
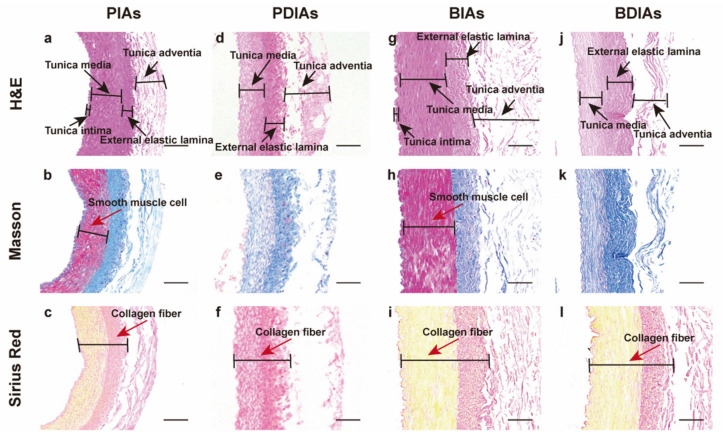
Characterizations of the DIAs. PIAs: porcine intercostal arteries; PDIAs: porcine decellularized intercostal arteries; BIAs: bovine intercostal arteries; and BDIAs: bovine decellularized intercostal arteries. H&E staining (**a**,**d**,**g**,**j**), Masson’s trichrome staining (**b**,**e**,**h**,**k**), and Sirius Red staining (**c**,**f**,**i**,**l**) of PIAs (**a**–**c**)/PDIAs (**d**–**f**)/BIAs (**g**–**i**)/BDIAs (**j**–**l**). Scale bar: 200 μm.

**Figure 4 bioengineering-11-00700-f004:**
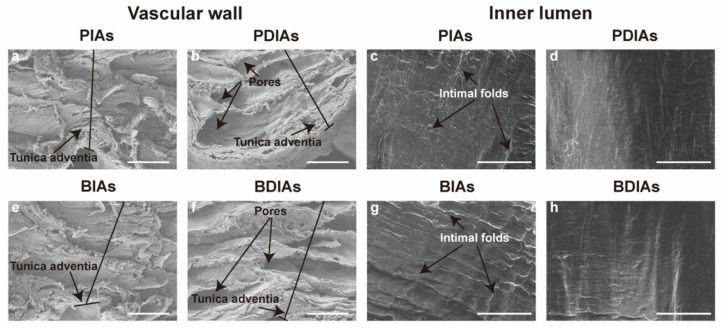
SEM images of the DIAs. PIAs: porcine intercostal arteries; PDIAs: porcine decellularized intercostal arteries; BIAs: bovine intercostal arteries; and BDIAs: bovine decellularized intercostal arteries. Microscopic observation of DIAs: inner lumen (**c**,**d**,**g**,**h**), scale bar: 10 μm; vascular wall (**a**,**b**,**e**,**f**), scale bar: 200 μm. Native vessels (**a**,**c**,**e**,**f**) and DIAs (**b**,**d**,**f**,**h**) of porcine (**a**–**d**) and bovine (**e**–**h**) origin.

**Figure 5 bioengineering-11-00700-f005:**
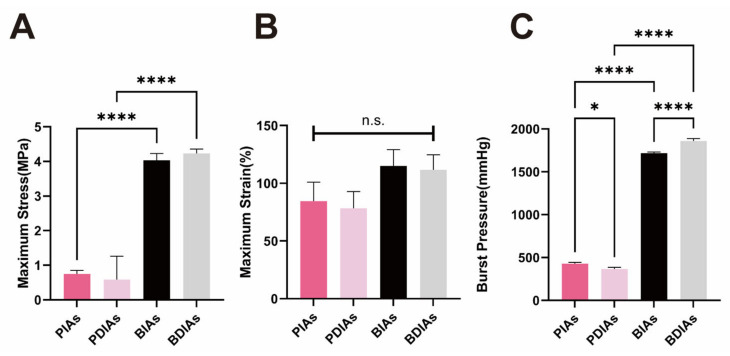
Mechanical characterizations of the DIAs. PIAs: porcine intercostal arteries; PDIAs: porcine decellularized intercostal arteries; BIAs: bovine intercostal arteries; and BDIAs: bovine decellularized intercostal arteries. Maximum stress (**A**), maximum strain (**B**), and burst pressure (**C**) for PIAs, PDIAs, BIAs, and BDIAs. Burst pressure was measured using a manometer; maximum stress and maximum strain were obtained using a tensile tester. n ≥ 3, * indicates *p* < 0.05, **** indicates *p* < 0.0001. “n.s.” indicates no significant differences between comparison groups.

**Figure 6 bioengineering-11-00700-f006:**
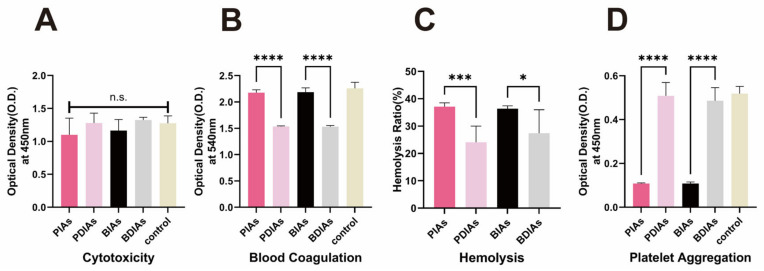
Biocompatibility and hemocompatibility of the DIAs. PIAs: porcine intercostal arteries; PDIAs: porcine decellularized intercostal arteries; BIAs: bovine intercostal arteries; and BDIAs: bovine decellularized intercostal arteries. (**A**) CCK-8 assay of the HUVECs cultured for three days, with and without medium supplemented with extract solution of each graft, respectively. “control” and “-” denote the same negative control group in which the HUVECs were cultured using normal medium. (**B**) Blood coagulation, (**C**) hemolysis, and (**D**) platelet aggregation for PIAs, PDIAs, BIAs, and BDIAs. The negative control consists of whole rabbit blood/diluted rabbit blood/resuspended platelet solution that did not come into contact with the material. n ≥ 6, * indicates *p* < 0.05, *** indicates *p* < 0.001, **** indicates *p* < 0.0001. “n.s.” indicates no significant differences between comparison groups.

## Data Availability

All data generated or analyzed during this study are included in this published article and its Appendix A. All further data will be provided by the corresponding author at any time upon request.

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
