# Peer review of "The Potential of a New Natural Vessel Source: Decellularized Intercostal Arteries as Sufficiently Long Small-Diameter Vascular Grafts"

_bioengineering, 2024, doi:10.3390/bioengineering11070700_

Round 1

Reviewer 1 Report

Comments and Suggestions for Authors

Excellent article in basic science with potential for wider applications in the field of coronary artery bypass surgery. 

steps of evaluation for cellularity, DNA content, collagen structure, bio-mechanical properties and histo-compatibilty are well documented with adequate statistical analysis with clear concise results. All illustrations are well organized and clearly convey the results.

Please add legends for abbreviations in each illustration.

appropriate adequate references

Reviewer 2 Report

Comments and Suggestions for Authors

In this manuscript, the authors explore the potential use of two types of decellularized intercostal arteries from xenogeneic large animals as new sources for coronary artery bypass grafts. They used two decellularization vessels and found that bovine intercostal arteries exhibit superior properties compared to porcine intercostal arteries, although both require further improvement in blood compatibility. This study identifies a suitably sized xenogeneic small-diameter artery and conducts preliminary investigations into its potential for human transplantation, which is somewhat novel. However, decellularized vessels are common in recent research, and the experimental content is simplistic. Here are the comments:

  1. While the authors provided some physical characterizations of the artificial vascular graft, the study lacks sufficient in vitro and in vivo experiments. For in vitro experiments, it is better to include recellularization experiments to see if ECs can attach to and grow well on the decellularized grafts. The study would be more convincing if the authors could transplant the vessels in animal models.
  2. Including one ex vivo or semi-ex vivo circulation experiment to verify acute thrombosis formation within 24 hours. For example, a temporary circulation between the rabbit carotid artery and jugular vein with the decellularized graft in between.
  3. In the Results section, please use consistent panel labels in sections 3.3, 3.4, and 3.5. Additionally, the results for each figure are not clearly explained.
  4. In 3.6, to test the hemocompatibility, the authors can provide SEM results to characterize the adhesion of blood cells.

Reviewer 3 Report

Comments and Suggestions for Authors

Please present in more detail the installation used for decellularization, in addition to the flow rate of the decellularization solution, it would also be useful to present the cannulation method, the perfusion pressure, the volume of the solution used for decellularization, possibly some serial photos with the evolution of the artery fragment in the bath decellularization.

Also, please, if possible, replace figure 1B with another one with a better resolution. It's hard to see the details. It would also be useful for the calibrated measuring instrument to be unitary in both images 1A and 1B.

Comments on the Quality of English Language

It would be useful for a native English speaker to review the manuscript from this point of view.

Round 2

Reviewer 2 Report

Comments and Suggestions for Authors

I have no more questions. I am looking forward to your future paper about the vascular grafts.